# Thermal Expansion of Plastics Used for 3D Printing

**DOI:** 10.3390/polym14153061

**Published:** 2022-07-28

**Authors:** Bruno Rădulescu, Andrei Marius Mihalache, Adelina Hrițuc, Mara Rădulescu, Laurențiu Slătineanu, Adriana Munteanu, Oana Dodun, Gheorghe Nagîț

**Affiliations:** 1Department of Digital Production Systems, “Gheorghe Asachi” Technical University of Iași, 700050 Iaşi, Romania; bruno.radulescu@academic.tuiasi.ro (B.R.); mara.radulescu@academic.tuiasi.ro (M.R.); adriana.munteanu@academic.tuiasi.ro (A.M.); 2Department of Machine Manufacturing Technology, “Gheorghe Asachi” Technical University of Iași, 700050 Iaşi, Romania; andrei.mihalache@tuiasi.ro (A.M.M.); slati@tcm.tuiasi.ro (L.S.); oanad@tcm.tuiasi.ro (O.D.); nagit@tcm.tuiasi.ro (G.N.)

**Keywords:** thermal expansion, experimental device, acrylonitrile butadiene styrene, polyethylene terephthalate, thermoplastic polyurethane, polylactic acid, experimental measurements, empirical mathematical models

## Abstract

The thermal properties of parts obtained by 3D printing from polymeric materials may be interesting in certain practical situations. One of these thermal properties is the ability of a material to expand as the temperature rises or shrink when the temperature drops. A test experiment device was designed based on the thermal expansion or negative thermal expansion of spiral test samples, made by 3D printing of polymeric materials to investigate the behavior of some polymeric materials in terms of thermal expansion or contraction. A spiral test sample was placed on an aluminum alloy plate in a spiral groove. A finite element modeling highlighted the possibility that areas of the plate and the spiral test sample have different temperatures, which means thermal expansions or contractions have different values in the spiral areas. A global experimental evaluation of four spiral test samples was made by 3D printing four distinct polymeric materials: styrene-butadiene acrylonitrile, polyethylene terephthalate, thermoplastic polyurethane, and polylactic acid, has been proposed. The mathematical processing of the experimental results using specialized software led to establishing empirical mathematical models valid for heating the test samples from −9 °C to 13 °C and cooling the test samples in temperature ranges between 70 °C and 30 °C, respectively. It was found that the negative thermal expansion has the highest values in the case of polyethylene terephthalate and the lowest in the case of thermoplastic polyurethane.

## 1. Introduction

Thermal expansion of solid materials is a property that considers the increase in size that characterizes the solid body as its temperature rises. The inverse property of thermal expansion is thermal contraction or negative thermal expansion.

Thermal expansion of a part is important in situations where this expansion could affect the integrity or behavior of the assembly of which the part is a component. Because the forces that occur during thermal expansion can be relatively large, they can cause deformation or even breakage of other parts or even the part affected by the expansion.

Thermal expansion is evaluated by considering the ratio between the increase in size under the action of increasing the temperature and the initial value of the dimension affected by heating. This ratio is the coefficient of thermal expansion and is usually used in the case of linear dimensions or volumes of parts affected by thermal expansion.

Significant differences exist between the expansion of metallic materials, plastics, or ceramics. It is appreciated that, in general, the thermal expansion of ceramic materials is less than the thermal expansion of metallic materials, as polymeric plastics have an expansion of about 10 times greater than those of metallic materials. Devices called dilatometers are used to evaluate linear thermal expansion.

With the increasing use of parts made of polymers by 3D printing, it has become important to know the thermal properties of polymers. These properties include thermal expansion.

Seven additive manufacturing technologies are currently considered: vat photopolymerization, material extrusion, material jetting, binder jetting, powder bed fusion, direct energy deposition, and sheet lamination [1]. The process used in this paper was material extrusion. This process involves a wire advanced and extruded through a heated nozzle and deposited in successive layers as a result of precisely controlled movements between the nozzle and the table of the 3D printer until the final part is obtained. Many input factors influence how successive layering occurs in the 3D printing process. These factors include the nozzle hole diameter, the relative movement speed between the nozzle and the printer table, the proposed level for the density of the workpiece material, the nozzle and printer table temperatures, and the cooling level during the 3D printing process, etc.). Due to the values of the input factors, the material of the part manufactured by 3D printing may have different degrees of densification and filling of the space inside the part, which will affect some thermal properties of the part material. Among these properties, there is thermal capacity expansion.

The need to consider thermal expansion as a property of the materials from which 3D printing processes made parts was highlighted by researchers [2,3,4,5]. Effective experimental attempts to study thermal expansion specific to polymeric materials used in manufacturing parts by 3D printing were performed by Wang [3] and Miller [6]. The importance of thermal expansion in the use of thermoactivated morphing materials has been highlighted by Nam [7]. The analysis of some thermal properties of the filaments used for 3D printing of parts of polymeric materials was developed by Trhlikova et al. [8] and Savu [9]. Zhang referred to the thermal expansion of power electronic components when he studied applications of additive manufacturing in this field [10].

The generation of 3D printed parts’ deviations due to thermal and negative thermal expansion was analyzed by researchers interested in a better knowledge of additive manufacturing processes, namely those of 3D printing [3,5,8,9].

Wang et al. [11] studied some characteristics of isotactic polypropylene. They found that the high crystallization rate of isotactic polypropylene can cause significant difficulties in the case of additive manufacturing processes. When studying the behavior of a composite material based on isotactic polypropylene, it was observed that the value of the coefficient of thermal expansion of isotactic polypropylene is about 10% higher than that of a composite material containing spray-dried cellulose nanofibrils.

Blanco considered thermal expansion when reviewing the applications of thermal analysis methods that can be used to manufacture parts by 3D printing processes [12]. He highlighted the availability of thermogravimetric analysis, differential scanning calorimetry, and dynamic mechanical analysis to assess the thermal expansion capacity of materials used to manufacture filaments when applying 3D printing processes.

In an overview of the additive manufacturing of polymer and polymer composite parts, Nat and Nilufar appreciated that the negative thermal expansion of parts after applying an additive manufacturing process induces residual stresses that can affect the behavior of parts during mechanical stresses [13]. They found a library created by Huang et al. [14] in connection with the possibilities of designing the parts so that the dimensions of the 3D printed part are affected as little as possible after the thermal shrinkage.

Zohdi et al. investigated the anisotropy of some thermal properties, as it is, among others, thermal expansion [15].

Hwang et al. have shown that a reduction in the value of the coefficient of thermal expansion is possible when copper particles are introduced into the ABS specimens [16]. For example, for a content of 50 wt.% of copper, the coefficient of thermal expansion decreases by about 29.5% compared to the value of the same coefficient valid for pure ABS, which means a decrease in the coefficient of thermal expansion from 108.2 ppm/°C to 76.2 ppm/°C. This way, the test samples’ significant deformation during thermal shrinkage is avoided.

The anisotropic character of the thermal expansion was highlighted by Baker et al. [17]. They compared the values of the coefficient of thermal expansion for test pieces printed along with longitudinal and transverse directions.

A relatively simple device has been proposed to allow direct observation of the spiral polymeric test samples during thermal or negative thermal expansion. This paper presents the research results to highlight the thermal expansion of spiral test parts made by the 3D printing of four different polymeric materials (styrene-butadiene acrylonitrile, polyethylene terephthalate, thermoplastic polyurethane, and polylactic acid). The experimental results were mathematically processed, and empirical mathematical models corresponding to the thermal contraction of the test samples were determined. 

## 2. Materials and Methods

### 2.1. Initial Considerations

In principle, it is considered that thermal expansion can be explained by the asymmetry of the potential energy, which determines an increase in the mean distance between atoms when the atoms vibrate along the line of interaction between them. However, non-vibrational contributions to the development of the thermal expansion process have also been highlighted [18].

When increasing the body temperature from a value *θ*_1_ to a value *θ*_2_, the magnitude Δ*L* of the thermal expansion of the body of length *L*_0_ is determined by using the relation:(1)$$\Delta L=\alpha L_{0}\left(\theta_{1}-\theta_{2}\right),$$
or:(2)$$\Delta L=\alpha L_{0}\Delta \theta ,$$
where *α* is the coefficient of linear thermal expansion, and Δ*θ* is the temperature variation.

Consistent with the above equation, the values of the coefficient of linear thermal expansion *α* correspond to the following relation:(3)$$\alpha =\frac{\Delta L}{L_{0}\Delta \theta }.$$

It is still possible to investigate the thermal expansion of the volume of a body. In this case, the volume thermal expansion coefficient *β* is valid:(4)$$\beta =\frac{\Delta V}{V_{0}\Delta \theta }.$$
where *V*_0_ is the initial volume of the body, and Δ*V*—the increase in body volume.

In most cases, the value of the linear expansion coefficient is important.

Devices called dilatometers, or dilatation analyzers, are used to determine the thermal expansion coefficients’ values experimentally. Pushrod dilatometers using optical interferometry or X-ray diffraction, optical dilatometers, capacitance dilatometers, etc., are known. A mention shall be made about horizontal and vertical dilatometers, respectively, concerning the position of a rod made of a material whose coefficient of thermal expansion is determined. In principle, when the temperature of a rod in the investigated material increases, the displacement of the edge of one end of the rod is determined using suitable optical means.

The expansion of the use of polymeric materials has led to the need to know some characteristics of the behavior of parts made of such materials to temperature variation. Manufacturers of polymeric materials generally provide brief information on the expansion properties of these materials. In industrial practice, there is sometimes the problem of more detailed knowledge of the thermal expansion of a body of polymeric materials at different temperatures.

On the other hand, the last decades have highlighted the possibilities of obtaining polymer parts of various shapes and sizes.

Under the conditions mentioned above and to design and materialize an easily accessible dilatometer, the idea of increasing the length of the bar in the material whose thermal expansion is of interest was formulated using a polymer test sample in the shape of an Archimedean spiral. It was accepted that, among the different categories of spirals, the Archimedean spiral ensures a maximum length of the test sample within a flat surface of predetermined dimensions.

Using a spiral-shaped test sample located in a spiral-shaped groove and allowing the spiral to elongate only in the direction of the outer end of the spiral test sample, the length of the test sample whose thermal expansion is to be measured could be increased (Figure 1). In this way, it would be possible to increase the accuracy of assessing the value of the linear thermal expansion coefficient due to considering the longer length of the test sample.

### 2.2. Experimental Conditions

Under the theoretical conditions mentioned above, a device has been designed to measure the thermal expansion of a spiral-shaped polymer test sample (Figure 1).

A groove in the shape of an Archimedean spiral was machined by milling on a numerically controlled machine tool [19]. An essential part of the device was an aluminum alloy plate. The outer end of the spiral groove is provided with a rectilinear segment. Inside the groove with a cross-section of 1.8 × 2.3 mm^2^, a polymeric test sample can be placed, having a shape corresponding to that of the groove in the aluminum alloy plate, providing a lateral gap of about 0.03 mm. A transparent material cover can be placed over the aluminum alloy plate and fixed to the plate with screws to avoid the external deformation of the test sample during thermal expansion.

The aluminum alloy from which the plate was made was of the EN-AW-2017 type AlCu4MgSi(A) alloy and contained 0.2–0.8% Si, 0.70% Fe, 3.5–4.5% Cu, 0.4–1.0% Mn, 0.4–1.0% Mg, 0.10% Cr, 0.25% Zn, 0.25% Ti, 0.05% other chemical elements and the rest—aluminum. As values of the main thermal properties of the aluminum alloy, it can mention solidification temperature 510 °C, melting temperature 645 °C, heat transfer capacity 873 J∙kg^−1^∙K^−1^, thermal expansion coefficient 22.9 µm∙m^−1^∙K^−1^, thermal conductivity 134 W∙m^−1^∙K^−1^.

The free end movement from outside the spiral test sample in the rectilinear segment could be highlighted using a dial gauge (Figure 1). However, it was appreciated that a higher accuracy would be obtained using an optical microscope. The microscope objective was fixed near the outer end of the linear segment of the test sample (Figure 2).

Regarding the proper way of measuring the variation of the length of the test sample, the cooling of the test sample in a freezer was first considered. The aluminum alloy plate, together with the test sample, could be removed from the freezer and placed in the working area of the optical microscope so that minute-by-minute measurements of the thermal expansion of the test sample could be performed. Therefore, the temperature of the aluminum alloy plate and the polymeric test sample could be measured using a gun-type infrared thermometer, which measures the infrared radiation emitted by a given surface. A non-contact infrared thermometer (manufactured by HOPPLINE—Hungary) was used for this purpose, with a measuring range between −50 °C and +400 °C (measuring distance of 5–15 cm, measuring accuracy of ±1.5 °C). 

The approximate distance from which the temperature was measured using the infrared thermometer was about 15 cm. Since the process of heat exchange between the aluminum alloy plate together with the spiral test piece and the external environment does not develop with the same intensity in all directions, the aluminum alloy plate temperature has different values at different points of the aluminum plate and, therefore, the average values of three measurements were considered.

The temperature evaluation of the aluminum plate was performed at 1-min intervals, determined by using a digital clock with a stopwatch (approximate time measurement accuracy of ±5 s). In the calculations, the average values of the measured temperatures were taken into account. As observed by using the finite element method and by measuring the temperature in different areas of the aluminum alloy plate, there are temperature differences, even when heated on the table of the 3D printer.

The following materials were considered as materials for the spiral test samples:(1)Black ABS (styrene-butadiene acrylonitrile). This material is used, for example, for the manufacture of general-purpose goods, such as toys, carcass parts, furniture, refrigerator interiors, helmets, etc.;(2)Red PET G (polyethylene terephthalate type G). Such material is used mainly in the textile industry, but also as a bottling or packaging material due to the lack of reaction with water or food;(3)Red color thermoplastic polyurethane (trade name: Flexfill 98A). This material has a hardness equal to that of rubber (98A). It is used especially in the manufacture of solid wheels for scooters and skateboards or in the manufacture of protective cases of smartphones;(4)Polylactic acid (type T PLA) silver color. This material is well adapted to the requirements of 3D printing processes. Its use in 3D printing does not require a heated substrate or a high melting temperature and increases the printing speed. It also ensures low manufacturing costs. A disadvantage is the increased sensitivity to the action of ultraviolet radiation. Parts with fine details can be easily made from polylactic acid. This material can be used to manufacture toys, jewelry, statues, etc.

The coefficient values of thermal expansion of these materials indicated in some specialized works can be observed in the second column of Table 1.

The thermal expansion of the polymer test sample from a temperature of about −9 °C (when it was removed from the freezer) to a temperature close to that of the environment in the laboratory, 13 °C) was thus measured using an optical microscope TM-1005B (manufactured by the Japanese company Mitutoyo. 

### 2.3. Evaluation of the Length of an Archimedean Spiral Test Sample

There are different ways to determine the length of an Archimedes spiral segment [22]. The length of the test sample in the form of an Archimedean spiral was calculated using the “Measure” function in the SolidWorks software. The lengths of the two side surfaces of the profile (with a rectangular cross-section, measuring 1.8 × 2.3 mm^2^) were measured. The arithmetic mean of the lengths of the two side surfaces was calculated, resulting in an average fiber length of 3337.63 mm. 

A problem with the device is that during the thermal expansion or contraction of the polymer spiral test sample, not only the spiral test sample expands or contracts but also the aluminum alloy plate in which the spiral groove the test sample is placed.

It is generally considered that the value of the coefficient of thermal expansion in the case of a metallic material is about ten times lower than that of a polymeric material. Specifically, in the case of aluminum alloy, the value indicated of the coefficient of thermal expansion is *α_Al_* = 22.9 m/(m∙K), while, for example, in the case of the PLA polymer is indicated *α*_PLA_ = 41 ∙ 10^−6^ m/m∙K, and in the case of the ABS polymer, *α*_ABS_ = 72 ∙ 10^−6^ m/(m∙K).

Suppose it takes into account that the size of the side of the square-shaped area of the aluminum alloy plate in which the spiral groove is located is about 128 mm, and assuming a temperature difference of 40 °C, this means a linear variation of the size of the aluminum alloy plate side:(5)$$\Delta L_{Al}=\alpha_{Al}L_{0\;Al}\Delta \theta =22.9\cdot 128\cdot 40=117.2\;µm.$$

Taking into account a length L ≈ 3337.63 mm of the spiral test sample in PLA and, respectively, in ABS and considering the values indicated for the coefficients of thermal expansion [20,21], it would be possible to increase the length of the spiral test sample Δ*L*_PLA_ = 5.47 mm and Δ*L*_ABS_ = 9.61 mm. It is found that the share *w* of the influence of the linear expansion of the aluminum alloy plate on the thermal expansion of the spiral test sample is very low (*w* = 0.1172 ∙ 100/5.47 = 2.14% in the case of the spiral test sample made of PLA and *w* = 0.1172 ∙ 100/9.61 = 1.21% in the case of the ABS spiral test sample, respectively. This situation allows us to neglect the influence exerted by the linear thermal expansion of the aluminum alloy plate on the thermal expansion of the spiral test sample of polymeric material. 

### 2.4. Simulation by the Finite Element Method of the Thermal Expansion of the Test Piece

The investigated process was modeled using the finite element method (FEM) and the ANSYS 3D software to evaluate the thermal expansion capacity of the spiral test samples of polymeric materials used in manufacturing parts by 3D printing. For the purpose mentioned above, the situation of a spiral test sample located in the spiral-shaped groove in the aluminum alloy plate was considered.

One of the results of finite element modeling is shown in Figure 3. The experimental tests will manage an uneven cooling or heating of the aluminum alloy plate. Due to heating or cooling at uneven speeds of different areas of the aluminum alloy plate where the polymer spiral test sample is located, there will be a distributed variation in temperature along with the spiral test sample. The value of the thermal expansion coefficient is variable with the temperature of the polymer spiral test sample. This means that the results will lead to an overall value of the evolution of the magnitude of the thermal expansion coefficient or negative thermal expansion coefficient.

It was developed a steady-state thermal FEM analysis considering the convection phenomena. Approximate convection coefficient values for aluminum alloys were confronted with those presented by Geng et al. [23]. It has conducted the analyses on ABS simulating the heat transfer for the cool environment, which starts from around −10 ° C and records about 13 °C on the top side of the cover plate (Table 2). The results are consistent with those measured by experimental means (Figure 3a). The FEM.-based purpose was to analyze further heat transfer on the spiral test sample by transferring steady-state thermal results into the static structural analysis. Results were plotted in the form of thermal strain, equivalent stress von Mises, equivalent elastic strain, and directional deformation in the direction of the heat flow. Thus, it may further visually assess the influence of heat transfer on various polymers. For example, it was clear that uneven cooling leads to a random distribution in the case of equivalent elastic strain (Figure 3b). The spiral undergoes multiple stages of deformation before it shrinks or expands in the designated groove. Static structural results highlight changes that occur to the original body. Directional deformation on the *Y*-axis gives approximate values to those obtained by experimental means. 

The initial conditions for steady-state thermal analysis include an initial ambient temperature of 22 °C and a temperature of −13.1 °C applied in increments on the lower surface of the plate, respectively. A process of free air convection of up to 2.5 W/mm^2^ °C applied to the upper surface of the plate was considered. The boundary conditions took into account both the contacts and the joints. The contacts are of three types, two of which include the frictionless thermal expansion of the 3D printed spiral and one that takes into account the friction between the spiral test sample and the groove walls. The joints refer to the spiral’s contact with the groove’s flat surface. Such conditions make it easier for the software to consider the material must flow only along the spiral groove.

The thermal imaging camera HT-18 (produced by Hti Instrument) was used to check the uneven heating of the aluminum plate on the table of the 3D printer. As it can be seen from Figure 3c, the image obtained by using the thermal imaging camera shows a real unevenness in the heating of the aluminum plate under the action of the table of the 3D printer. This uneven heating of the aluminum plate results in an uneven distribution of the spiral test sample temperature.

## 3. Results

The results obtained in the experimental tests regarding the thermal expansion of the test samples made of the four materials are presented in Table 2. A graphical representation of the Δ*L* increase in the lengths of the test samples made of polymeric materials when the temperature increases from approximately −9 °C to +13 °C and corresponding to the experimental results in Table 2 can be observed in Figure 4.

Later, the idea of measuring the linear negative thermal expansion of the test samples appeared. The heating of the aluminum alloy plate on the table of a 3D printer was considered (Figure 5). In such a case, the temperature of the upper area of the table can be programmed, this being an input factor in the 3D printing process. Thus, measurements of the negative thermal expansion were performed when the test sample cooled, from a temperature of 70 °C to the ambient temperature in the laboratory (30 °C), during the measurements. The measurement results of the test samples shrinkage are included in Table 3. By considering the results in Table 3, a graphical representation of the evolution of negative thermal expansion over time was made (Figure 6).

The results of measurements on the thermal expansion of the previously polymer spiral test sample in the freezer can be seen in Table 2. In contrast, the results of the thermal shrinkage of the polymer spiral test sample previously heated on the table of a 3D printer were included in Table 3.

The values indicated in Table 4 were determined by comparing the maximum values of thermal expansion and negative thermal expansion, respectively, at the temperature variation intervals in the case of experimental research.

For the experimental results in Table 4, which show more pronounced differences in terms of negative thermal expansion for the four materials used in the experimental research, mathematical processing was used by means of specialized software [24], to determine empirical mathematical models able to highlight the influence of temperature on linear negative thermal expansion.

The specialized software was generated by taking into account the least squares method. It provides conditions for selecting the most appropriate empirical mathematical model from five such models: the first-degree polynomial, the second-degree polynomial, the power function, the exponential function, and the hyperbolic function. For the selection, the value of the so-called Gauss criterion was used. This value (the Gauss’s criterion value) is defined using the sum of the squares of the differences between the ordinate’s values determined by the empirical mathematical model considered and the values determined experimentally. The lower the value of the Gauss criterion, the more appropriate the mathematical model is to the experimental results used.

The most appropriate empirical mathematical models determined using specialized software, and the models determined were included in the second column of Table 5.

## 4. Discussion

In many situations in manufacturing engineering, empirical mathematical models of the power function type have been determined and used. Such empirical mathematical models are sometimes used to determine the influence of various factors on the cutting speed, the size of the components of the cutting forces, the size of a roughness parameter of the machined surface, etc. [25,26,27]. An advantage of using empirical mathematical models of power function type derives from the possibility of evaluating the intensity of the influence exerted by a certain factor by analyzing the value of the exponent attached to that factor in the function of power type to the values of exponents attached to other factors. However, empirical mathematical models of the power function type are particularly appropriate when dealing with monotonous evolutions of the output parameter as the value of the input factor increases or decreases in the investigated process. Such a situation, i.e., the lack of minimums or maximums, was confirmed, for example, by the graphical representation in Figure 5, where there is a continuous increase in thermal expansion when the temperature increases by heating the spiral test samples. For this reason, power function empirical mathematical models for the four polymeric materials were also determined, and those models can be seen in the second column of Table 5.

By considering the values of temperatures and thermal expansions or negative thermal expansions in Table 2 and Table 3, it was possible to determine the values of the coefficients of thermal expansion corresponding to each of the four materials used to make the spiral test samples. Some aspects of calculating the coefficients of thermal expansion coefficients starting from the experimental values and Equation (3) are presented in Table 5.

The graphical representations in Figure 4 and Figure 6 have been elaborated considering the experimental results in Table 2 and Table 3. In the case of the graphical representation in Figure 3, there is a certain reversal of the thermal expansion values in the cases of ABS and PLA polymeric materials for the period from the beginning of heating. Subsequently, with the development of the expansion process, it is found that the thermal expansion of PLA exceeds that of ABS.

A clearer highlight of the differences between the negative thermal expansion of the spiral test samples in the two materials is provided by the content of the graphical representation in Figure 7, which confirms the order defined by the results recorded for longer time intervals than those in the case of the diagram in Figure 4.

The most appropriate empirical mathematical models for the experimental results were used to draw the graphical representation in Figure 7. The descending order of the intensity of the influence exerted by the cooling process on the negative thermal expansion can be noticed in this diagram, where polymeric materials are arranged, from this point of view, in the order: polyethylene terephthalate (PET)—polylactic acid (PLA)—styrene-butadiene acrylonitrile (ABS)—thermoplastic polyurethane (Flexfill).

As seen from Table 3, at the beginning of the experiment, heating the aluminum alloy plate and the test samples on the table of the 3D printer, was made on a temperature of 70 °C.

To reduce the friction between the spiral test samples and the spiral groove walls, a thin layer of mineral oil-based lubricant was applied to the support plate before the insertion of the test samples, both when applying the thermal expansion or negative thermal expansion.

In a cross-section, the profile of the spiral test samples made of the four materials has the shape of a rectangle with sides of 1.8 mm and 2.3 mm. The test sample width was 1.8 mm (value measured at a temperature of 22 °C). After the insertion of the spiral test sample in the groove with the shape of an Archimedean spiral, the lateral clearance between the test sample and the groove was about 0.03 mm, at a temperature of 22 °C. 

Comparing the maximum values of thermal and negative thermal expansion, respectively, at the temperature variation intervals in the case of experimental research, we reach the values indicated in Table 4. 

Comparing the results obtained experimentally with the values indicated in other works shows that the determined coefficients of thermal expansion were lower (Figure 8).

Possible explanations for this could be the following:-There are, of course, differences between the properties of a part made by 3D printing and the properties of the bulk material. Due to the characteristics of the 3D printing manufacturing process, the material density of the 3D printing test sample may be much lower than in the case of bulk material. This could be the main cause of the relatively large differences between the values of the coefficients of linear thermal expansion indicated by the materials manufacturers or identified in specialized documents and the values determined experimentally using the proposed equipment. It should be noted that the values of the coefficients of thermal expansion determined experimentally were lower than those of the coefficients of linear expansion indicated by the manufacturers of the materials or identified in specialized documents in the case of all four materials;-The thermal expansion way of the spiral sample is different from that of a strictly linear test sample;-The relatively small gap between the spiral test sample and the spiral groove in the aluminum alloy plate could lead to frictional forces along the relatively long length of the spiral test sample, and these forces diminish the free expansion of the test sample;-The existence of roughness resulting from the spiral groove as a result of the spiral groove generation by milling could also contribute to a braking of the free thermal expansion;-A possible gap at the end of the test sample inside the spiral groove could also reduce the thermal expansion or the negative thermal expansion measured at the end of the segment in the form of a straight line of the spiral test sample.

A certain adjustment of the density of the material of the test sample made by 3D printing is possible by acting on a size that provides some information about the printing process and is called “infill”. It must still be taken into account that the density of the test sample material is not uniform, being higher near the outer walls and lower inside the test sample, where there may even be gaps of different sizes. There are, moreover, other print parameters of the printer whose values can change the density of the printed material and therefore can affect the values of the coefficient of thermal expansion. As such, it is expected that there will be a difference between the values of thermal expansion determined using a dilatometer and the proposed equipment, respectively, as long as the dilatometer indicates the value of the linear thermal expansion and the proposed equipment—the thermal expansion along a flat spiral.

Such issues may be further examined in the future to determine their influence on the final results on the values of the coefficients of expansion.

References to the coefficient of linear thermal expansion exist in ASTM E831, ASTM D696, and ISO 11359. These documents describe the test procedure valid for thermomechanical analysis when aiming at linear expansion of the test sample. As this paper aims to expand along a spiral and uses the equipment for such testing, only a few of the provisions included in the above standards could be met, including considering the equipment available for conducting experimental research.

Some aspects that may lead to differences between the results of determining the coefficient of thermal expansion using the proposed equipment and a dilatometer, respectively, have been briefly presented above. In the case of thermomechanical analysis and the use of a dilatometer, the test sample must be 12.7 mm (0.5″) wide and 75 mm (3″) long. The end surfaces of the test sample must be flat. Together with the support on which it is placed, the test sample is introduced into heating equipment and a gradual increase in temperature takes place, with a predetermined heating rate (for example, 10 °C/min) and in a temperature range default (for example, from −30–+30 °C). This ensures conditions for a controlled and as uniform heating as possible of the test sample. In the case of the proposed equipment, simple and accessible heating sources were used, but which do not allow slow and uniform heating of the spiral test sample. Additionally, the existence of a gap between the spiral test sample and the groove in the aluminum plate could affect to some extent the final result, as the thermal expansion of the aluminum plate exerts a certain influence on the value of the determined coefficient of expansion.

## 5. Conclusions

For different practical situations, it may be necessary to know the ability of materials used for 3D printing of parts to expand as the temperature rises, and how they shrink when the temperature decreases. A physical quantity used to evaluate this property is the coefficient of thermal expansion. The possibility of determining the size of the coefficient of thermal expansion in the case of four materials used for the manufacture of parts by 3D printing was considered. The materials called acrylonitrile butadiene styrene, polyethylene terephthalate, thermoplastic polyurethane, and polylactic acid were taken into consideration. An experimental device was designed based on an aluminum alloy plate, in which a spiral groove was made by milling. Test samples of the four polymeric materials manufactured by 3D printing were placed in this spiral groove. It was considered that using a spiral test sample would allow an increase in the length of the test sample whose thermal expansion is measured and could provide additional information on the process of thermal expanding or negative thermal expanding of the test sample. Finite element modeling of the heating of the spiral test sample in the spiral groove in the aluminum alloy plate has led to the observation that there is a variation in temperature along the spiral due to cooling at different speeds of the aluminum alloy plate spiral groove is located. In the case of experimental research, the expansion measurement of the outer end of a rectilinear segment of the test sample was performed using an optical microscope. The experimental results were mathematically processed using specialized software based on the least squares method. The specialized software allowed the selection of the most appropriate empirical mathematical models from five mathematical models frequently used in experimental research of different processes in manufacturing technology. The graphical representations elaborated by considering the experimental results and the empirical mathematical models determined by the experimental results showed that, among the materials analyzed, the most intense negative thermal expansion is obtained in the case of polyethylene terephthalate. At the same time, the lowest heat shrinkage was noticed in the case of thermoplastic polyurethane material. In the future, it is intended to expand the experimental research on test samples made of other polymeric materials and deepen the research related to the non-uniform material heating of the test sample in the aluminum alloy plate. Experimental research will also be carried out on changing the thermal expansion or negative thermal expansion coefficients with temperature variation.

## Figures and Tables

**Figure 1 polymers-14-03061-f001:**
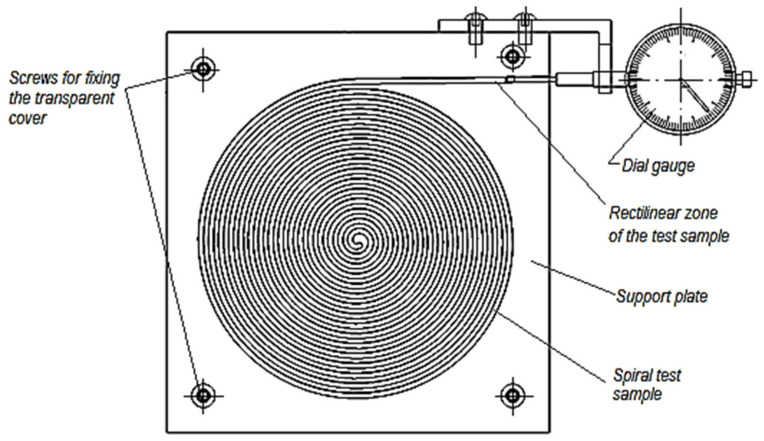
Schematic representation of the device for measuring the thermal expansion of a polymer spiral test sample in the variant involving a dial gauge.

**Figure 2 polymers-14-03061-f002:**
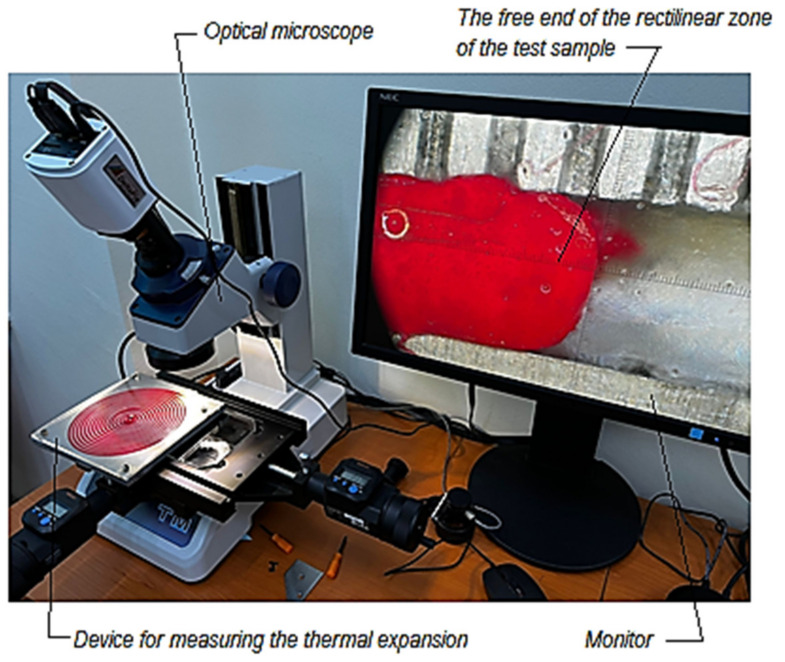
Image of the device for measuring the thermal or negative thermal expansion of the spiral test sample made of polymeric material Flexifill 98A, in the variant involving an optical microscope.

**Figure 3 polymers-14-03061-f003:**
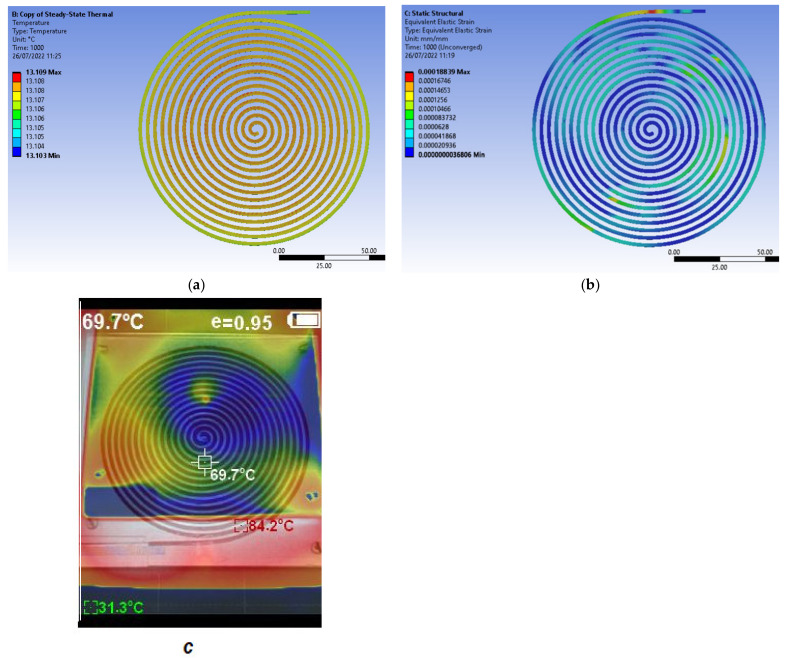
Uneven cooling/heating of the spiral test sample due to different heat transfer rates in distinct areas of the aluminum alloy plate: (**a**) temperature distribution in the steady-state thermal scenario; (**b**) equivalent elastic strain distribution in the static-structural scenario; (**c**) image obtained using the thermal imaging camera.

**Figure 4 polymers-14-03061-f004:**
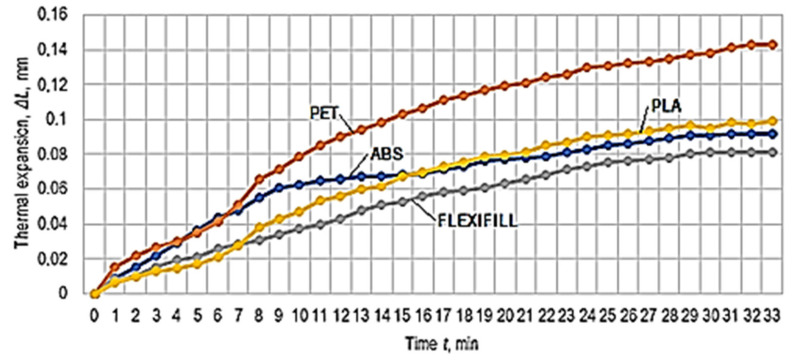
The evolution of thermal expansion over time, with small differences between the initial temperatures.

**Figure 5 polymers-14-03061-f005:**
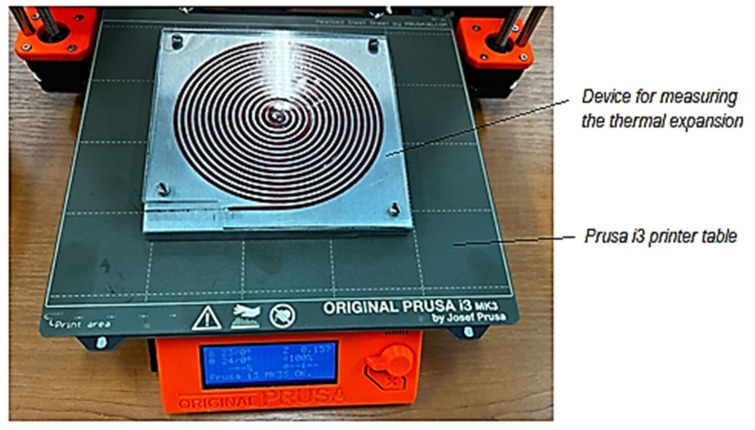
Image of a device for measuring the expansion or thermal shrinkage of a polymer spiral test sample together with the aluminum alloy plate placed on the printer’s table to reach the temperature at which the spiral test sample begins to cool and contract.

**Figure 6 polymers-14-03061-f006:**
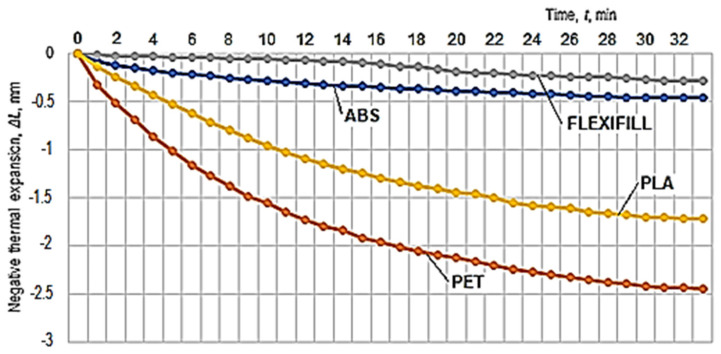
The evolution in time of negative thermal expansion of some spiral specimens made of polymeric materials, there being small differences between the initial temperatures.

**Figure 7 polymers-14-03061-f007:**
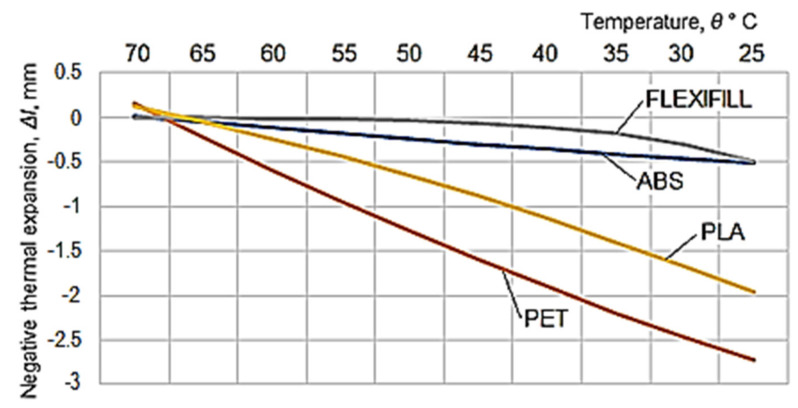
Evolution of negative thermal expansion to the initial temperature, according to the most appropriate empirical mathematical models to the experimental results, for the four polymeric materials.

**Figure 8 polymers-14-03061-f008:**
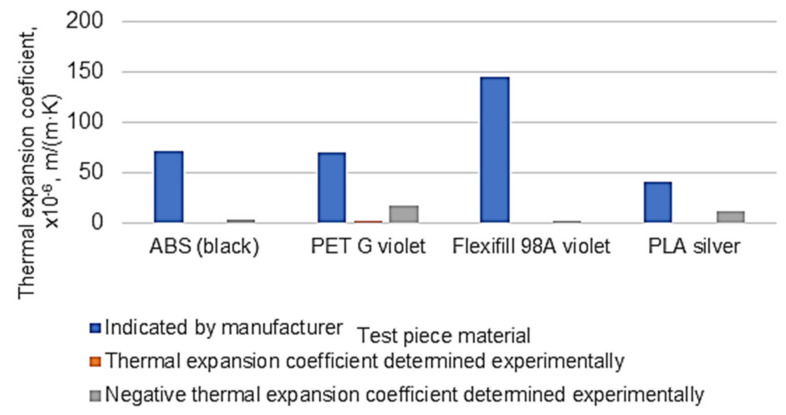
Highlighting the differences between the values of the coefficients of linear thermal expansion indicated by the material manufacturers or identified in specialized documents [20,21] and the values obtained in case of thermal expansion of the spiral test sample.

**Table 1 polymers-14-03061-t001:** Values of thermal expansion coefficients for materials used in experimental research.

Polymeric Material	Values of the Coefficient of Thermal Expansion Indicated in Specialized Documents, in m/(m·K) [20,21]
ABS (black)	72 ∙ 10^−6^
PET G (violet)	7 ∙ 10^−5^
Flexfill 98A (red)	145 ∙ 10^−6^ (approximate value, determined by comparison with those of similar materials)
PLA (silver)	41 ∙ 10^−6^

**Table 2 polymers-14-03061-t002:** Experimental results regarding the thermal expansion of some polymeric materials.

Time, min	ABS Black		PET G Violet		FLEXIFILL 98A Red		PLA Silver	
	Temperature, °C	Thermal Expansion, mm	Temperature, °C	Thermal Expansion, mm	Temperature, °C	Thermal Expansion, mm	Temperature, °C	Thermal Expansion, mm
0	−8.8	0	−11.5	0	−8.3	0	−9.2	0
1	−2.6	−0.009	−2.5	−0.015	−5.7	−0.008	−6.3	−0.006235
2	−2.3	−0.015	−2.3	−0.022	−3.9	−0.01	−3.7	−0.009896
3	−0.6	−0.022	−1.3	−0.027	−3.4	−0.015	−2.5	−0.012831
4	−0.6	−0.029	−0.5	−0.03	−1.2	−0.019	−1.1	−0.0147
5	1.3	−0.036	0	−0.035	−0	−0.021	−0.1	−0.017148
6	2.2	−0.044	1.7	−0.041	−0.1	−0.026	−0.9	−0.021195
7	2.4	−0.048	2.7	−0.051	1.8	−0.028	1.7	−0.027816
8	4	−0.055	3.4	−0.066	3	−0.031	3.2	−0.037932
9	5	−0.061	4.7	−0.071	3.9	−0.034	4.3	−0.042731
10	6	−0.062	5.2	−0.079	4.1	−0.037	4.8	−0.046699
11	6.5	−0.065	5.5	−0.085	4.6	−0.04	5.5	−0.053651
12	6.8	−0.066	6.5	−0.09	5.7	−0.043	6.3	−0.056203
13	7.4	−0.067	7.2	−0.094	6.5	−0.048	7.4	−0.060043
14	8.2	−0.067	7.5	−0.098	6.8	−0.051	7.9	−0.06196
15	8.3	−0.068	7.9	−0.103	7.2	−0.053	8.2	−0.067081
16	9.1	−0.069	8.3	−0.106	7.9	−0.056	8.7	−0.069362
17	9.3	−0.071	8.8	−0.111	8.5	−0.058	9.2	−0.073242
18	9.8	−0.073	9.5	−0.114	9.2	−0.059	9.7	−0.075759
19	10.1	−0.076	10	−0.117	9.9	−0.061	10.2	−0.078333
20	10.6	−0.077	10.2	−0.119	10.4	−0.063	10.6	−0.079781
21	10.9	−0.078	10.5	−0.121	10.8	−0.066	10.9	−0.08126
22	11	−0.079	10.8	−0.124	11.1	−0.068	11.2	−0.085492
23	11.5	−0.081	11	−0.126	11.4	−0.071	11.5	−0.086936
24	11.9	−0.083	11.3	−0.13	11.6	−0.073	11.8	−0.090123
25	12	−0.085	11.9	−0.131	12.1	−0.075	12.1	−0.090997
26	12.1	−0.086	12	−0.132	12.3	−0.076	12.3	−0.091983
27	12.3	−0.088	12.2	−0.133	12.5	−0.077	12.4	−0.093028
28	12.4	−0.089	12.4	−0.135	12.6	−0.078	12.6	−0.094724
29	12.5	−0.091	12.5	−0.137	12.7	−0.08	12.7	−0.096244
30	12.7	−0.091	12.6	−0.138	12.9	−0.081	12.9	−0.094936
31	12.8	−0.092	12.9	−0.141	13.1	−0.081	13.2	−0.09782
32	12.9	−0.092	13.1	−0.143	13.2	−0.081	13.3	−0.09745
33	13.1	−0.092	13.2	−0.143	13.2	−0.081	13.4	−0.098847

**Table 3 polymers-14-03061-t003:** Experimental results regarding the negative thermal expansion of some polymeric materials.

Time, min	ABS Black		PET G Violet		FLEXIFILL 98A Red		PLA Silver	
	Temperature, °C	Negative Thermal Expansion, mm	Temperature, °C	Negative Thermal Expansion, mm	Temperature, °C	Negative Thermal Expansion, mm	Temperature, °C	Negative Thermal Expansion, mm
0	69.7	0	70	0	71.8	0	71	0
1	64.9	−0.08	62	−0.32	66.7	−0.01	61	−0.132
2	61.5	−0.118	60	−0.514	63.8	−0.022	57.7	−0.24
3	58.8	−0.147	57.2	−0.685	59	−0.028	56.2	−0.332
4	56.8	−0.174	55.6	−0.86	55.8	−0.032	53	−0.432
5	54.7	−0.197	54.2	−1.015	54.9	−0.035	51.6	−0.527
6	52.6	−0.216	52	−1.164	52	−0.039	49.5	−0.617
7	51.3	−0.235	50.2	−1.273	51.4	−0.044	48	−0.71
8	50.2	−0.254	48.8	−1.382	50.3	−0.05	46.5	−0.797
9	47.9	−0.27	47.1	−1.481	48.5	−0.054	45.2	−0.88
10	47	−0.285	45.4	−1.556	45.7	−0.057	43.9	−0.964
11	45.3	−0.297	44.5	−1.652	44.5	−0.064	42.7	−1.03
12	44.4	−0.308	43.4	−1.729	42.5	−0.073	41.6	−1.091
13	43.8	−0.321	42.2	−1.798	41.8	−0.081	40.4	−1.149
14	42.2	−0.335	41	−1.844	40.7	−0.088	39.7	−1.197
15	41.6	−0.343	40.4	−1.92	39.1	−0.097	38.6	−1.238
16	40.7	−0.354	39.3	−1.956	38.6	−0.11	37.6	−1.296
17	39.7	−0.364	38.6	−2.013	38.2	−0.13	37.1	−1.336
18	38.6	−0.372	37.8	−2.053	37.7	−0.14	36.3	−1.376
19	38	−0.381	37	−2.088	36.9	−0.16	35.7	−1.412
20	37	−0.391	36.3	−2.128	36.2	−0.19	35.1	−1.44
21	36.7	−0.396	35.6	−2.166	35.4	−0.2	34.9	−1.462
22	35.8	−0.408	34.9	−2.209	35.1	−0.21	34.5	−1.501
23	35.1	−0.412	34.3	−2.242	34.5	−0.22	33.5	−1.553
24	34.8	−0.42	33.7	−2.267	33.9	−0.23	32.8	−1.582
25	33.8	−0.426	33.1	−2.293	33.3	−0.23	32	−1.592
26	33.6	−0.434	32.8	−2.323	32.8	−0.24	31.3	−1.614
27	33.2	−0.441	32.5	−2.357	32.4	−0.24	30.1	−1.642
28	32.7	−0.448	32	−2.378	31.9	−0.25	29.2	−1.663
29	32.2	−0.454	31.6	−2.398	31.5	−0.26	28.7	−1.679
30	32	−0.461	30.8	−2.424	30.9	−0.27	28.5	−1.698
31	31.5	−0.463	30.2	−2.432	30.4	−0.28	28.4	−1.702
32	31.1	−0.464	29.5	−2.438	29.8	−0.28	28.2	−1.711
33	30.4	−0.465	28.8	−2.441	29.3	−0.29	27.9	−1.712

**Table 4 polymers-14-03061-t004:** Calculated values of the thermal and negative thermal expansion coefficients, considering the experimental results presented in Table 2 and Table 3.

Material	On Heating			On Cooling		
	The Maximum Value of the Thermal Expansion, Δ*L*, mm	Temperature Variation, Δ*θ*, K	Value of the Coefficient of Thermal Expansion *α*	The Maximum Value of Negative Thermal Expansion, Δ*L*, mm	Temperature Variation, Δ*θ*, K	Value of the Coefficient of Thermal Negative Expansion *α*
Column no. 1	2	3	4	5	6	7
ABS (black)	0.092	8.8 + 13.1 = 21.9	1.25 ∙ 10^−6^	0.465	69.7 − 30.4 = 39.3	3.54 ∙ 10^−6^
PET G violet	0.143	11.5 + 13.2 = 24.7	1.73 ∙ 10^−6^	2.441	70 − 28.8 = 41.2	17.7510^−6^
Flexifill 98A violet	0.081	8.3 + 19.2 = 21.5	1.12 ∙ 10^−6^	0.290	71.8 − 29.3 = 42.5	2.04 ∙ 10^−6^
PLA silver	0.098	9.2 + 13.4 =22.6	1.31 ∙ 10^−6^	1.712	71 − 27.9 = 43.1	11.90 ∙ 10^−6^

**Table 5 polymers-14-03061-t005:** Empirical mathematical models designed to highlight the negative thermal expansion to the temperature decrease, considering the experimental results.

**Polymer**	**The Most Appropriate Empirical Mathematical Model for the Experimental Results and the *S_G_* Value of Gauss’s Criterion**	**Empirical Mathematical Model of the Power Function Type and the *S_G_* Value of Gauss’s Criterion**
Black ABS	Second-degree polynomial Δ*L* = −0.739 + 0.00757*θ* + 0.0000457*θ*^2^ *S_G_ *= 1.078667 ∙ 10^−4^	Δ*L* = 3074565*θ*^−3.76^ *S_G_ *= 1.840823 ∙ 10^−2^
Violet PET G	Second degree polynomial Δ*L* = 3.805−0.0356*θ*−0.000298*θ*^2^ *S_G_ *= 2.356737 ∙ 10^−3^	Δ*L* = 1672831*θ*^−3.789^ *S_G_ *= 0.7223308
Red FLEXIFILL 98A	Exponential function Δ*L* = 6.281 ∙ 0.904*^θ^* *S_G_ *= 2.715475 ∙ 10^−4^	Δ*L* = 11457241*θ*^−4.468^ *S_G_ *= 1.078153 ∙ 10^−3^
Silver PLA	Second-degree polynomial Δ*L* = 3.658−0.0753*θ* + 0.000305*θ*^2^ *S_G_ *= 3.64125 ∙ 10^−3^	Δ*L* = 2463873*θ*^−4.062^ *S_G_ *= 0.4127897

## Data Availability

Data are available on request from the authors.

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
