# Peer review of "Thermal Expansion of Plastics Used for 3D Printing"

_polymers, 2022, doi:10.3390/polym14153061_

Round 1

Reviewer 1 Report

The authors proposed a method to evaluate the thermal expansion of polymers using a spiral test sample. The experimental activities described in this paper were not explained, even if the results could be easily reproduced in other labs. The temperature range of the analysis between -13 and 20 °C does not encompass the natural function of 3D printed products. 

Moreover, the scientific robustness of the paper is low because

- the introduction is too broad and does not give sufficient insight into the problem.

- no comparisons were made with the traditional technique using a dilatometer.

- the results achieved with this method were lower than those generally reported in the literature.

- the English language should be massively improved. Several errors and mismatches were found in the text.

- no comparisons were made with the traditional technique using a dilatometer.

- the results achieved with this method were lower than those reported in the literature.

Author Response

Authors’ responses to the reviewers' comments

The authors of the reviewed paper wish to express their gratitude for the efforts of the reviewers invested in the analysis of the proposed paper and for the useful observations and suggestions to improve the quality of the paper.

All the changes were highlighted in the manuscript of the paper by using the color green.

REVIEWER 1

Reviewer's comment no. 1. The introduction is too broad and does not give sufficient insight into the problem.

Authors response to the reviewer's comment. We respect the reviewer's opinion, but the article's introduction proposed for publication has a length of 61 pages or about 1.15 pages. From the web page https://www.mdpi.com/journal/polymers, it is found that the work Atmakuri, A.; Palevicius, A.; Vilkauskas, A.; Janusas, G. Numerical and Experimental Analysis of Mechanical Properties of Natural-Fiber-Reinforced Hybrid Polymer Composites and the Effect on Matrix Material. Polymers 2022, 14, 2612. https://doi.org/10.3390/ polym14132612 has an introduction of almost 3 pages. Many articles published in the journal Polymers have longer introductions than the proposed article.

Reviewer's comment no. 2. No comparisons were made with the traditional technique using a dilatometer

Authors response to the reviewer's comment. The authors appreciate that the reviewer is right. The paper presented the values ​​of the thermal expansion coefficients indicated by the companies producing materials for 3D printing. A more detailed comparison should consider that the material in the printed parts is not evenly distributed and even has gaps. It should also be noted that, for example, in a cross-section through the spiral specimen, there are at least two areas of the cross-section, the one printed in the first phase on the printer table and the one printed towards the end of printing, which have different densities of the material inside the test sample. Moreover, the operator or programmer of the 3D printer has the possibility to change a size called “infill”, with a significant influence on the structure and density of the test sample material and, therefore on its coefficient of expansion. In these circumstances, a comparison with the results of an evaluation using a dilatometer (which measures a linear expansion and not one along a spiral) does not seem to lead to the identification of new explanations, all the more so as, as mentioned above, the values ​​of the coefficients of expansion indicated by the manufacturers of the test material were included.

Explanations of this kind were also introduced in the text of the manuscript.

For this reason, an infilling coefficient of 100% was used to generate all the spirals, in an attempt to diminish the possibility that a part of the expansion or contraction of the material will be absorbed by the degree of infilling of the spiral test sample. As a consequence, the material layers of 3D printing of these spirals generate almost a “full” part. The shape of filament deposited that generates a layer from each material on 3D printing has an approximate elliptical cross-section, so we can say that between successive layers, even if the print is 100%, there will always be microspaces resulting from this deposition way.

Another parameter kept constant during the realization of all the spirals, regards the diameter of the extruded filament from which the spirals were made. In this case, it is Ø 0.2 mm.

Another parameter used at the same values for 3D printing of specimens of all materials was the speed of the print head. For all materials it was the same.

Reviewer's comment no. 3. The results achieved with this method were lower than those generally reported in the literature.

Authors response to the reviewer's comment. The reviewer is right. Some additional explanations for this were included in the answer to the previous question.

Even if a lubricant was used to reduce friction, the coefficient of friction could still be of some importance in the investigated process.

Reviewer's comment no. 4. The English language should be massively improved. Several errors and mismatches were found in the text.

Authors response to the reviewer's comment. The reviewer is right. The authors tried to make a correction of the manuscript text in accordance with the rules of expression in English.

Reviewer's comment no. 5. No comparisons were made with the traditional technique using a dilatometer.

Authors response to the reviewer's comment. The observation repeats the content of comment no. 2. There is the authors' response to that comment.

Reviewer's comment no. 6. Authors response to the reviewer's comment. The observation repeats the content of comment no. 3. There is the authors' response to that comment.

Reviewer 2 Report

General comment:

The article  investigates the thermal expansion properties of the 3d printing parts. the paper is well structured but can be further improved by addressing the following concersn.

specific comments:

1. how many specimens were tested for each material?

2. please provide the average value and standard deviation on the results

Author Response

Authors’ responses to the reviewers' comments

The authors of the reviewed paper wish to express their gratitude for the efforts of the reviewers invested in the analysis of the proposed paper and for the useful observations and suggestions to improve the quality of the paper.

All the changes were highlighted in the manuscript of the paper by using the color green.

REVIEWER 2

Reviewer's comment no. 1. How many specimens were tested for each material?.

Authors response to the reviewer's comment. Only one test sample of each material was tested, aiming only to differentiate the materials by taking into account their thermal expansions.

Reviewer's comment no. 2. Please provide the average value and standard deviation of the results.

Since we considered sets corresponding to distinct points belonging to 4 distinct curves, determining an average value would not provide significant information. The mean value and the standard deviation have significant values in the case of a set of tests performed under absolutely the same conditions, or here we followed the variation of thermal expansion as a function of time.

Reviewer 3 Report

 -Citations should be more specific. Citing [1-10] in one place doesn’t convey much meaningful information.

-There are different categories of additive manufacturing processes. The authors need to specify the relevant processes to this study and discuss how different processes may have an effect on the thermal properties.

-Are there any ASTM or ISO standards relevant to determining thermal expansion?

-L206, it should be Table 1, not Table 3.

-It would be more meaningful to present length vs. temperature (Fig. 3 and Fig. 5).

-How the exact temperature is determined at the time of length measurement needs to be described more clearly.

-How did the authors control the temperature of the sample also needs to be described.

-Some contents in Sec. 2 can be moved to the Results section.

-It is also interesting to see thermal expansion coefficients vs. temperature in a line graph.

-The boundary/initial conditions of the FE model need to be described more clearly.

-Given that the temperature isn’t very uniform over the entire spiral plate, how did the authors account for that?

-Texts in Fig.6 a, and b are difficult to read.

-The image quality of Fig. 7 is poor.

-More discussion comparing the authors’ results and results from the literature should be included. A bar chart would be good to visualize this.

-Discuss the difference between the 3D printed material and bulk material properties.

-Paragraphs in the Discussion section read fragmented and can be better organized.

-Are any of the experiments repeated? There’s no error bar presented.

Author Response

Authors’ responses to the reviewers' comments

The authors of the reviewed paper wish to express their gratitude for the efforts of the reviewers invested in the analysis of the proposed paper and for the useful observations and suggestions to improve the quality of the paper.

All the changes were highlighted in the manuscript of the paper by using the color green.

REVIEWER 3

 Reviewer's comment no. 1. Citations should be more specific. Citing [1-10] in one place doesn’t convey much meaningful information.

Authors response to the reviewer's comment. We considered the reviewer to be right and reformulated the citation.

Reviewer's comment no. 2. There are different categories of additive manufacturing processes. The authors need to specify the relevant processes to this study and discuss how different processes may have an effect on the thermal properties

Authors response to the reviewer's comment. The authors considered that the reviewer was right. Information has been introduced in relation to the issues mentioned above.

Reviewer's comment no. 3 Are there any ASTM or ISO standards relevant to determining thermal expansion?

Authors response to the reviewer's comment. References to the coefficient of linear thermal expansion exist in ASTM E831, ASTM D696 and ISO 11359. These documents describe the test procedure valid for thermomechanical analysis, when aiming at linear expansion of the test sample. As this paper aims to expand along a spiral and use of equipment for such testing, only a few of the provisions included in the standards mentioned above could be met, including by considering the equipment available for conducting experimental research.

A text containing the previous information has been included in the revised version of the manuscript.

Reviewer's comment no. 4. L206, it should be Table 1, not Table 3

Authors response to the reviewer's comment. The reviewer is right.

Reviewer's comment no. 5. It would be more meaningful to present length vs. temperature (Fig. 3 and Fig. 5)

Authors response to the reviewer's comment. The suggestion made by the reviewer is interesting and the authors thank the reviewer for the proposal, but at the moment the application of the suggestion would require the resumption of experimental tests. For this reason, the authors considered as useful in supporting their arguments the graphical representations of thermal expansion vs. time.

The graphical representation in figure 7 corresponds to a certain extent to the request of the reviewer.

Reviewer's comment no. 6. How the exact temperature is determined at the time of length measurement needs to be described more clearly.

Authors response to the reviewer's comment. The time variation of thermal expansion was recorded using a stopwatch. It is estimated that the time assessment was made with an accuracy of ± 5 s. A statement to this effect was included in the text of the paper.

Reviewer's comment no. 7. How did the authors control the temperature of the sample also needs to be described.

Authors response to the reviewer's comment. The temperature of the specimen was measured using the infrared thermometer, at a distance between the thermometer of approximately 15 cm. As mentioned, the plate containing the spiral groove and the spiral test sample was heated using the mass of a 3D printer or cooled in a freezer. Due to an uneven heat dissipation from the aluminum plate to the outside environment, however, the temperature was not the same throughout the plate and this was also highlighted by the finite element modeling of the temperature distribution in the aluminum plate.

A brief explanation in this regard has been introduced in the text of the manuscript.

The temperature of the test sample was controlled using an infrared pyrometer with a temperature range of -50 O C … 400 O C. The measurement accuracy was ± 1.5 O C.

Reviewer's comment no. 8. Some contents in Sec. 2 can be moved to the Results section.

Authors response to the reviewer's comment. We considered the reviewer to be right and repositioned some of the text and figures from Section 2 in the Results section.

Reviewer's comment no. 9. It is also interesting to see thermal expansion coefficients vs. temperature in a line graph.

Authors response to the reviewer's comment. The authors appreciate that the reviewer's suggestion is interesting, but could not be realized in the relatively short time given to improve the initial quality of the article. A response to the reviewer's suggestion was included in the response to comment no. 5 of the reviewer.

Reviewer's comment no. 10. The boundary/initial conditions of the FE model need to be described more clearly.

Authors response to the reviewer's comment. The initial conditions for steady-state thermal analysis include an initial ambient temperature of 22 o C and a temperature of -13.1 o C applied in increments on the lower surface of the plate, respectively. A process of free air convection of up to 2.5 W/mm2 o C applied to the upper surface of the plate was considered. The boundary conditions took into account both the contacts and the joints. The contacts are of three types, two of which include the frictionless thermal expansion of the 3D printed spiral and one that takes into account the friction between the spiral test sample and the groove walls. The joints refer to the contact of the spiral with the flat surface of the groove. Such conditions make it easier for the software to take into account the fact that the material must flow only along the spiral groove.

Reviewer's comment no. 11. Given that the temperature isn’t very uniform over the entire spiral plate, how did the authors account for that?

Authors response to the reviewer's comment. The ambient temperature was not strictly controlled. Thus, as soon as the heat or cooling source stops working, the materials react differently. The same type of behavior was predicted by FEM analyzes. As mentioned above, the temperature values in the tables are average values of the results of three measurements.

As can be seen in the image taken with the thermal imaging camera or in the FEM analysis, there are several models that we can see in both situations. It is true that the outside of the test sample cools/heats up faster than the inside of the test sample.

Reviewer's comment no. 12. Texts in Fig.6 a, and b are difficult to read

Authors response to the reviewer's comment. In order to improve the reading conditions, an enlargement of the components of the figure was resorted to.

Reviewer's comment no. 13. The image quality of Fig. 7 is poor.

Authors response to the reviewer's comment. Figure 7 has been redone.

Reviewer's comment no. 14. More discussion comparing the authors’ results and results from the literature should be included. A bar chart would be good to visualize this

Authors response to the reviewer's comment. A bar chart was introduced in the paper (Figure 8. Highlighting the differences between the values of the coefficients of linear thermal expansion and the values obtained in the case of thermal expansion of the spiral test sample).

Reviewer's comment no. 15. Discuss the difference between the 3D printed material and bulk material properties.

Authors response to the reviewer's comment. Further explanations of the differences between the values of coefficients of thermal expansion indicated by the manufacturers of materials used for 3D printing or identified in specialized documents and the values of coefficients of expansion of the spiral test sample obtained experimentally using the proposed equipment were introduced in the article.

Reviewer's comment no. 16. Paragraphs in the Discussion section read fragmented and can be better organized.

Authors response to the reviewer's comment. An attempt was made to remedy the situation reported by the reviewer.

Reviewer's comment no. 17. Are any of the experiments repeated? There’s no error bar presented.

Authors response to the reviewer's comment.

So far, the experiments have not been repeated. As shown in the response to the reviewer's comment no. 14, a bar chart was introduced in the paper

Reviewer 4 Report

1. Table 3 needs to be renamed as Table 1 on page 6.

2. Figures 5 and 6 should be reformatted to ensure that the Time axis values do not overlap with the data points or the plots.  Also get rid of the grid lines.

3. In Table 4, fix the typo in the Temperature variation, Δθ, K for PLA Silver.

4. Include more references where the Gauss criterion has been employed for similar applications under Results (page 12).

5. What studies are being proposed to be done in the future to understand why there is a significant difference between the experimental values and the values from literature for the thermal expansion coefficients?

6. Please proof read the article and fix typos and syntax errors.

Author Response

Authors’ responses to the reviewers' comments

The authors of the reviewed paper wish to express their gratitude for the efforts of the reviewers invested in the analysis of the proposed paper and for the useful observations and suggestions to improve the quality of the paper.

All the changes were highlighted in the manuscript of the paper by using the color green.

REVIEWER 4

Reviewer's comment no. 1. Table 3 needs to be renamed as Table 1 on page 6.

Authors response to the reviewer's comment. The reviewer is right. Table 3 on page 6 has been renamed Table 1.

Reviewer's comment no. 2. Figures 5 and 6 should be reformatted to ensure that the Time axis values do not overlap with the data points or the plots.  Also get rid of the grid lines.

Authors response to the reviewer's comment. Figures 5 and 7 have been redone. However, we maintained the grid lines, in order to be able to more easily follow the information in the figures.

Reviewer's comment no. 3. In Table 4, fix the typo in the Temperature variation, Δθ, K for PLA Silver.

Authors response to the reviewer's comment.. We consider that we have made the correction requested by the reviewer.

Reviewer's comment no. 4. Include more references where the Gauss criterion has been employed for similar applications under Results (page 12).

Authors response to the reviewer's comment. To identify an empirical mathematical model to match the experimental results, a specialized software was used, developed by G. CreÅ£u (reference [24] in the proposed article). This software uses the least squares method to identify different types of models (first degree polynomial, second degree polynomial, power type function, exponential function, etc.) and Gauss’s concordance criterion to determine which is the most more appropriate empirical mathematical model among the several available mathematical models. It is true that, at present, the coefficient of determination R2 is mostly used to evaluate the concordance of the empirical model with the experimental results.

Information on Gauss's concordance criterion is relatively little on the web (for example, Anghelache C., Radu, I. Construction of the regression model for the economic risk analysis. Romanian Journal of Statistics - Supplement no. 7/2020, p. 23-42, available at https://www.revistadestatistica.ro/suplement/wp-content/uploads/2020/07/rrss_07_2020_a1_en.pdf Page 39 of this article gives the calculation expression of the Gaussian concordance criterion ).

Considering that the results obtained using the mentioned software are valid, the authors of the reviewed article used this software in several previously published articles.

A reference to the use of Gauss's criterion for choosing the most appropriate empirical mathematical model from several such available models has been identified in a somewhat old book (Worthing, A. G., Geffner, J. Treatment of experimental data. New York, John Willey & Sons, Inc. 1943 (we have a Romanian translation of the book since 1959.) In subchapter IX.14 of this work, it is shown that (retranslation from Romanian into English): “Gauss's criterion says that the equation that best fits is the one that makes the minimum ratio between the sum of the squares of the deviations of the values obtained from the calculated values of y and the number of observed value pairs (x, y) minus the number of arbitrary constants involved. n represents the number of pairs of observed values and m - the number of arbitrary constants or free parameters, the criterion is written:

(96)”

As the subject of the article does not, however, refer to the basics of the specialized software for processing experimental results, no additional references have been introduced in the reference list of the proposed article.

Reviewer's comment no. 5 What studies are being proposed to be done in the future to understand why there is a significant difference between the experimental values and the values from literature for the thermal expansion coefficients?.

Authors response to the reviewer's comment. As there are many factors corresponding to the 3D printing process, it is intended to design and materialize a factorial experiment, highlighting the direction and intensity of the influence of input factors in the 3D printing process on thermal expansion, including different materials. to make the spirals.

Reviewer's comment no. 6. Please proof read the article and fix typos and syntax errors..

Authors response to the reviewer's comment. The reviewer is right. The text of the article has been re-verified.

Round 2

Reviewer 1 Report

The article was revised and this version is complete.

Some minor improvements are needed for English language and  paper formatting (e.g. line 246-261)

Author Response

Please see the document attached.

Reviewer 2 Report

The replies are satisfactory. 

Author Response

Please see the document attached.

Reviewer 3 Report

The authors have addressed most comments. At the end of the discussion, can the authors add some comments on how departing from existing standards may affect the measurement result. Some formating and layout of figures may be fixed before final publication.

Author Response

 Reviewer's comment no. 1. The authors have addressed most comments. At the end of the discussion, can the authors add some comments on how departing from existing standards may affect the measurement result. Some formating and layout of figures may be fixed before final publication

Authors response to the reviewer's comment. We appreciated that the reviewer was right and included additional comments.
